# Prevalence and Distribution of Three Bumblebee Pathogens from the Czech Republic

**DOI:** 10.3390/insects13121121

**Published:** 2022-12-05

**Authors:** Alena Votavová, Oldřich Trněný, Jana Staveníková, Magdaléna Dybová, Jan Brus, Olga Komzáková

**Affiliations:** 1Agricultural Research Ltd., Troubsko, Zahradní 1, 664 41 Troubsko, Czech Republic; 2Department of Geoinformatics, Faculty of Science, Palacký University Olomouc, 17. Listopadu 50, 771 46 Olomouc, Czech Republic

**Keywords:** bumblebees, pathogen, Crithida bombi, Apicystis bombi, Nosema bombi

## Abstract

**Simple Summary:**

In recent decades, there has been a significant global decline in pollinators. In addition to the honey bee, it is bumblebees that contribute significantly to the pollination of many wild and farm plants. Many factors, such as habitat loss, climate change, pesticides and pathogens, are contributing to the decline of bumblebee populations. The main parasites of bumblebees include *Crithidia bombi*, *Apicystis bombi* and *Nosema bombi*. In our study, we aimed to obtain the first knowledge of their occurrence in the Czech Republic in the two most abundant bumblebee species (buff-tailed bumblebee *Bombus terrestris* and red-tailed bumblebee *Bombus lapidarius*). More than half of the captured buff-tailed bumblebee individuals were infected by *C. bombi*, less than a quarter of *N. bombi* individuals and the least by *A. bombi*. Red-tailed bumblebee individuals were infected less frequently. Surprisingly, more individuals were infected with all three parasite species than with only the combination of *N. bombi* and *A. bombi*. Parasite infection is also influenced by the environment. More individuals were infected with *C. bombi* in urban and forested landscapes than in grasslands and agricultural landscapes. In turn, a higher incidence of *N. bombi* was found around greenhouses where commercially produced bumblebees were used.

**Abstract:**

Bumblebees are significant pollinators for both wild plants and economically important crops. Due to the worldwide decrease in pollinators, it is crucial to monitor the prevalence and distribution of bumblebee pathogens. Field-caught bumblebee workers and males were examined for the presence of three pathogens during the summer months of the years 2015–2020 (*Bombus terrestris/lucorum*) and 2015–2017 (*Bombus lapidarius*). The greatest prevalence was in the case of *Crithidia bombi*, where significantly more workers (57%) of *B. terrestris/lucorum* were infected than males (41%). Infection was also confirmed in 37% of *B. lapidarius* workers. The average prevalence was very similar in the case of *Nosema bombi* in workers (25%) and males (22%) of *B. terrestris/lucorum*. In the case of *B. lapidarius*, 17% of the workers were infected. The lowest number of infected individuals was for *Apicystis bombi* and the prevalence of infection was significantly higher for males (22%) than workers (8%) of *B. terrestris/lucorum*. Only 3% of workers and 4% of males of *B. terrestris/lucorum* were simultaneously infected with three types of pathogens, but no worker was infected with only a combination of *N. bombi* and *A. bombi*. The greatest prevalence of *C. bombi* was found in urban or woodland areas.

## 1. Introduction

Bumblebees are important pollinators of many wild plants and agricultural crops [1,2]. This makes evidence of the decrease or elimination of pollinators in many countries worldwide alarming. In Europe, over 20% of bumblebees are at risk of extinction, and populations are declining in nearly 50% of species [3]. The primary determining causes of their worldwide decline include climate change, habitat loss, pesticides and pathogens [4,5,6]. Apart from viruses, the most common bumblebee pathogens influencing their population dynamics, behaviour and ecology are *Crithidia bombi*, *Nosema bombi* and *Apicystis bombi*.

The intestinal trypanosome *C. bombi* is one of the most common pathogens in the bumblebee *Bombus terrestris*. Vertical transmission takes place either via infected nest material or infected nestmates [7,8]. Horizontal transmission presumably takes place via blossoms infected with faeces [9]. *C. bombi* infections can cause changes in bumblebee behaviour, as well as problems distinguishing between blossoms with nectar and empty flowers, diminishing their ability to forage for food [10]. It can also reduce the survival of bumblebee queens during hibernation [11] and their ability to successfully establish a nest [12]. The microsporidian *N. bombi* is the second most common bumblebee pathogen. Transmission usually takes place by ingestion of spores from infected food, but the infection can also be transmitted during mating [13]. *N. bombi* infections negatively influence nest size and the lifespan of queens, workers and drones [14,15,16]. A significant role in the prevalence of this illness is played by climatic conditions and host genotype [17,18,19]. *N. bombi* is considered to be a significant pathogen in the context of worldwide bumblebee decline, whose prevalence is also influenced by the reported spillovers across host species and between commercial and natural bumblebee populations [20,21]. The sporozoan *A. bombi* is a neogregarinid pathogen of bumblebees and honeybees. Transmission takes place by ingestion of oocysts, i.e., the oral-faecal route of transmission, just as with the previous pathogens. Sporozoites subsequently develop in the gut and are then deposited in the fat body. Infected queens have a reduced fat body, which also diminishes their ability to survive hibernation [22,23]. Workers are also negatively influenced by fat body reductions, as this is the place where a number of biochemical reactions take place, which are important for immunity and metabolism [24]. Recently it has been considered a serious cause of bumblebee decline in South America, where it was apparently introduced together with commercially bred bumblebees [25].

The prevalence of these three pathogens has already been monitored in various bumblebee species in a number of countries [26,27,28,29,30,31,32], but no data on the incidence of these pathogens in the Czech Republic has been published. Here, we provide an assessment of the health condition of the bumblebee population and findings on the prevalence of pathogens in *Bombus terrestris/lucorum* and *Bombus lapidarius* in the Czech Republic.

## 2. Materials and Methods

The collection of *B. terrestris/lucorum* species individuals (workers and males) took place between the years 2015–2020, and individuals of *B. lapidarius* (workers and males) in the years 2015–2017, each time in July and August. Specimens were preserved in 96% ethanol and stored at a temperature of −20 °C.

For testing, we used DNA isolated from the entire internal contents of the abdomen of the studied individuals using the commercial kit DNasy Blood & Tissue Kit^®^ QIAGEN (Hilden, Germany). Parasitic DNA was detected using a PCR diagnostic method with primers amplifying part 18S of rDNA specific for the studied parasites. To detect the *C. bombi* pathogen, we used primers Crith-F/R (623 bp amlicon) according to Schmid-Hempel & Tognazzo (2010) [33], for the detection of *A. bombi* we used ApBF1/ApUR2 (258 bp amplicon) according to Meeus et al. (2010) [34] and for the detection of *N. bombi* we used Nbombi-SSU-Jfl/Jrl (323 bp amplicon) according to Klee et al. (2006) [35]. Reactions were carried out on the final reaction volume of 25 µL and contained 1 µL of template DNA, 12.5 µL 2 × PPP Master Mix (Top-Bio, Vestec, Czech Republic), 1 µL of each of the primers (10 µM) and ultrapure water were added to the final volume of the reaction. Reactions were carried out using the Thermal Cycler TC-512 device (Techne, Duxford Cambrige, UK) under the following conditions: 95 °C 4 min; 28 × (94 °C 20 s; 50 °C 20 s; 68 °C 35 s); 72 °C 5 min.

The PCR reaction included a positive and negative control. For the positive control, we used the DNA of the infected individual from which the product was PCR sequenced using the LIGHTRUN service (GATC Biotech, Ebersberg, Germany), the acquired sequence was then compared using the BLAST algorithm with the sequence of the corresponding pathogen in the public NCBI database, and it was confirmed that the sequence belonged to the given pathogen. Quality control of isolated DNA and results of the PCR reaction were analysed using horizontal electrophoresis in 2% agarose gel with Midori Green^®^ (NIPPON Genetics EUROPE, Düren, Germany). On the Tvrdonice locality, there are greenhouses with commercially produced bumblebees.

Maps of sampling locations were produced using ArcGIS Pro 3.0.2 (Redlands, CA, USA) [36]. Landscape type was determined with the help of the ArcGIS Pro 3.0.2 [36] using the CORINE Land Cover 2018 [37] map based on the geographic coordinates recorded during capture. Individual types were merged into four categories: urban (urban development, areas of urban greenery, sports and recreational spaces, industrial or commercial zones), agricultural landscape (orchards, vineyards, arable land, agricultural areas with natural vegetation), meadowland (meadows, pastures) and woodland (deciduous, coniferous and mixed forest).

The dependency between pathogen infection and caste/landscape type was tested using Pearson’s Chi-square test using R v. 4.2.1.

## 3. Results

### 3.1. Bombus terrestris/lucorum

A total of 540 individual specimens of *B. terrestris/lucorum* were collected from 126 locations (Appendix A). Of these, an infection by the pathogens *C. bombi*, *A. bombi* or *N. bombi* was detected in 396 workers and 144 males (Table 1).

Year-on-year, *C. bombi* infection (Figure 1) was between 49–70% for workers and 29–53% for males (due to the low number of males, the years 2015–2017 were not evaluated). On average (years 2015–2020), *C. bombi* infections were confirmed in 57% of workers and 41% of males. There is a statistically significant difference between male and worker infection rates (*p* < 0.001).

A difference between infected males and workers (*p* < 0.001) was also found for *A. bombi* infections (years 2015–2020), where 8% of workers and 22% of males were infected on average. Year-on-year, the recorded infection prevalence (Figure 2) was between 3–17% in workers and 9–42% in males (due to the low number of males, the years 2015–2017 were not evaluated).

Infections with the *N. bombi* pathogen (Figure 3) were confirmed in an average of 25% of workers and 22% of males. Year-on-year, recorded infections were between 9–48% in workers and 21–24% in males (due to the low number of males, the years 2015–2017 were not evaluated). Unlike with *C. bombi* and *A. bombi*, in the case of *N. bombi,* no difference between castes was discovered (*p*-value 0.32; years 2015–2020). Only in 2018 was the infection rate significantly higher for males than for workers (*p* < 0.05).

In 13 workers and six males of *B. terrestris/lucorum*, infection by all three types of pathogens was discovered. For infections by two types of pathogens, the highest incidence was in the CV (*C. bombi* and *N. bombi*) group (in 61 workers and in 10 males). Conversely, no joint infection was found in workers in the AV (*A. bombi* and *N. bombi*) group (Figure 4).

Landscape type and pathogen infection differ significantly for only *C. bombi* (*p*-value 0.013); for *A. bombi* and *N. bombi* they are not significantly different, with *p*-values of 0.67 and 0.27, respectively (Figure 5). Contingency table residuals suggest there are more *C. bombi* infections in urban and woodland environments in contrast to agricultural and meadowland types.

In 2017, sampling was carried out in the locality of Tvrdonice, which contained greenhouses utilising commercially sold bumblebees. Individuals caught at this location were significantly more likely to be infected with the *N. bombi* pathogen (*p*-value 0.008) compared to the other assessed locations. No difference in infection rate was proven for the other pathogens. Even when comparing locations from other years, where over 10 individuals were captured, the incidence of *N. bombi* at the Tvrdonice location was the highest (Appendix A).

### 3.2. Bombus lapidarius

A total of 88 individuals of *B. lapidarius* were collected from 17 locations (Appendix A). Of these, 81 were workers, and seven were males (Table 2). In *B. lapidarius* workers, an average of 38% of individuals were infected with *C. bombi*, 4% with *A. bombi* and 12% were *N. bombi*. The year-on-year prevalence of worker infections was 31–50% for *C. bombi*, 3–6% for *A. bombi* and 3–38% for *N. bombi*. For males, there was only a low number of collected individuals in 2016. Of the seven individuals collected, *C. bombi* infection was confirmed in three individuals, *A. bombi* in 1 individual and *N. bombi* also in one individual.

Joint infection with all three types of pathogens was only recorded in three males. In five workers and one male, joint infection with *C. bombi* and *N. bombi* was shown (Figure 6).

We did not plot the presence of pathogens by landscape type due to low to no samplings in most landscape types.

### 3.3. Comparison of B. terrestris/lucorum and B. lapidarius

A chi-square test confirmed the significantly lower prevalence of *C. bombi* (*p* < 0.001) and *N. bombi* (*p*-value 0.013) for workers of *B. lapidarius* in comparison with *B. terrestris/lucorum* workers.

## 4. Discussion

Our study is the first to look at the incidence and prevalence of three types of pathogens in *B. terrestris/lucorum* and *B. lapidarius* in the territory of the Czech Republic. A total of 132 locations were visited during mapping.

The highest percentage of average infections out of the three monitored pathogens was recorded for *C. bombi*, in both *B. terrestris/lucorum* workers (57%) as well as *B. lapidarius* workers (37%). Earlier findings show that the prevalence of *C. bombi* changes during the season. While in June, it reaches relatively low values (15%, or 19% for *B. lapidarius*), in July in *B. terrestris,* it can rise as high as 78% (or 65% in *B. lapidarius*). In August, the prevalence then gradually falls to 58% (or 63% in *B. lapidarius*) [27]. These results also match our findings, which for *B. terrestris/lucorum* correspond more to the August prevalence. In the case of *B. lapidarius*, a third of all samples were collected at the start of July, in other words, at the beginning of the culmination of *C. bombi* prevalence in the bumblebee population. For *B. terrestris/lucorum* males, the average prevalence of *C. bombi* was lower (41%) than in workers, which corresponds with the findings of Shykoff and Schmid-Hempel (1991) [26], who explain the difference by the higher likelihood of infection within the nest. While males tend to leave the nest within 4 days of hatching, workers spend much more time in the nest, and the risk of infection is, therefore, higher. Apart from this, workers visit more blossoms over their lifetime, which also increases their risk of infection from infected blossoms [9].

The recorded infections of *A. bombi* in *B. terrestris/lucorum* was around 3–42%, with a significantly higher infection rate in males (22%) than in workers (8%). The different prevalence between castes in *B. terrestris/lucorum* was also observed in the south of Spain [30]. For *B. lapidarius* workers, 4% of the collected individuals were infected.

A significant role in the prevalence of *N. bombi* is played by climatic conditions and host genotype [17,18,19]. *N. bombi* infections in *B. terrestris/lucorum* fluctuated year-on-year between 9–48%, which is consistent with the observations of Manlik et al. (2017) [17]. Previous studies report a higher prevalence and abundance of *N. bombi* in males than in workers [26,38,39]. Nevertheless, in our study, the average prevalence of *N. bombi* did not differ between castes and was conversely very similar in both males (22%) and workers (25%) of *B. terrestris/lucorum*. A significant difference between castes was only found in 2018, where 9% of workers were infected alongside 21% of males. In *B. lapidarius* workers, the percentage of infections was between 3–38%, year-on-year. On average, they were less often infected (17%) than workers of *B. terrestris/lucorum*.

As has been previously mentioned, *N. bombi* is considered to be a significant pathogen in the context of the worldwide bumblebee decline, whose prevalence is also influenced by the reported spillovers across host species and between commercial and natural bumblebee populations [20,21]. In 2017, we, therefore, collected 37 samples of *B. terrestris/lucorum* in the Tvrdonice location, the site of greenhouses with commercially produced bumblebees. At this location, we recorded a significantly higher incidence of *N. bombi* (62%) than at the other locations (25%) that year. Even when comparing 10 locations from subsequent years, where over 10 individuals were captured, the incidence of *N. bombi* at the Tvrdonice location was the highest. The prevalence of *C. bombi* also increased; however, the increase was not significant. In the case of *A. bombi*, the detection was low at that location and did not differ from the other locations. It is interesting that Graystock et al. (2014) [28] in England conversely found a higher prevalence of *C. bombi* and *A. bombi* in the vicinity of greenhouses, while detection of *N. bombi* was very low (less than 1%). They also, however, had a low detection rate of *N. bombi* at locations far from greenhouses.

Our results also confirm that, in a small percentage of cases, bumblebees can be simultaneously infected with multiple types of pathogens, which is in line with the findings of Graystock et al. (2014) [28]. The highest number of simultaneous infections (15%) was found in *B. terrestris/lucorum* workers, which is also the case of the CV (*C. bombi* and *N. bombi*) combination. The CA (*C. bombi* and *A. bombi*) combination then infected 3% of workers and 6% of males. Conversely, the AV (A. *bombi* and N. *bombi*) combination was not discovered in any *B. terrestris/lucorum* workers, nor in any individual specimen of *B. lapidarius* and was only found in 2 males of *B. terrestris/lucorum*. What is interesting is that despite no detection of simultaneous infection in the AV group of pathogens, 3% of workers and 4% of males of *B. terrestris/lucorum* were infected with all three pathogen types (CAV). The likelihood of detecting the simultaneous infection by two pathogens should, however, be many times higher than the likelihood of detecting three pathogens in a single individual. As we know, mixed infections may change the impact of individual pathogens on disease symptoms [40]. For example, Graystock et al. (2015) [23] found that Deformed wing virus infections in bumblebees could mitigate the sub-lethal effect (degreased sugar sensitivity, intermediate lipid:body mass ratio) of *A. bombi*. Viruses were not included in our study, but the question remains whether or not the CAV combination may perhaps be ultimately less lethal than the AV combination on its own. Nevertheless, this hypothesis requires further research.

Habitat type may have a non-negligible effect on pathogen prevalence. Bee density, differences in the type and number of available flowers or exposure to environmental stressors may be driving factors. A recent study carried out on the honeybee in the Czech Republic found an increased number of individuals infected with *Crithidia mellificae* in urban landscapes over agricultural ecosystems [41]. In our study, for *B. terrestris/lucorum* we also found a significantly higher difference in the prevalence of *C. bombi* in urban landscapes in comparison with meadowlands and field vegetation, which is in line with previous findings [5,42]. A higher population density and environmental stress may be good explanations for the difference. Nevertheless, in our study, we also found a higher prevalence of *C. bombi* infections in woodland habitats. The individuals in our study were almost exclusively captured on flowers. While it was relatively difficult to find flowering vegetation allowing for specimen capture in forests and towns during the late summer months, collection in field vegetation was also largely carried out in wide clover fields and flowering meadows, which provided sufficient food even late in summer and the number of individuals at these locations tended to be relatively high. Sufficient food may reduce the number of visits made by different individuals to each blossom, which can lead to a reduction in horizontal transmission via infected blossoms and mitigate food scarcity stress, in contrast to the limited food sources in forests and towns, where a larger number of individuals are competing for a single food source. Without a doubt, the factors influencing pathogen prevalence are much more complex and require further research.

## Figures and Tables

**Figure 1 insects-13-01121-f001:**
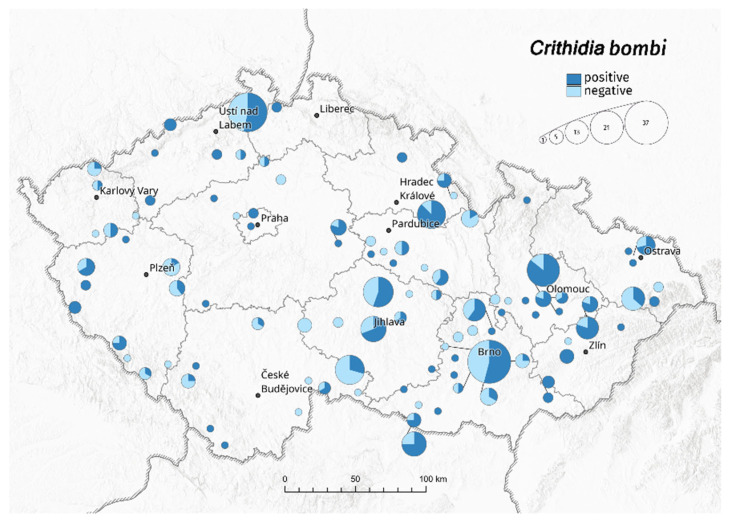
Sampling locations and the ratio of *Crithidia bombi* infection to uninfected bumblebees in the Czech Republic. The circles reflect the number of samples in sampling locations.

**Figure 2 insects-13-01121-f002:**
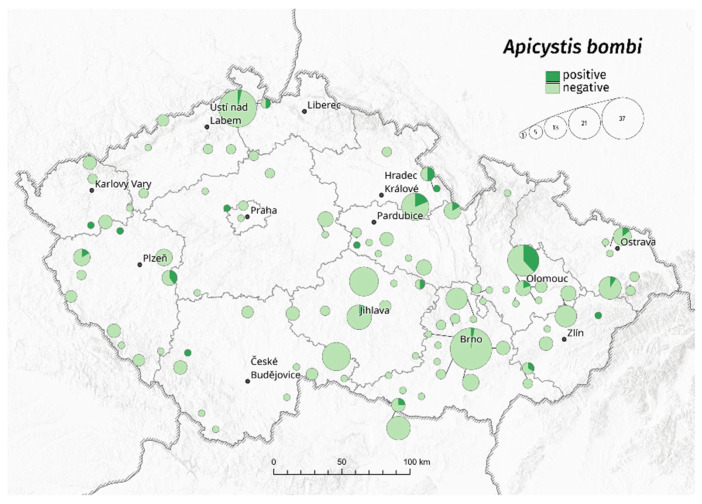
Sampling locations and the ratio of *Apicystis bombi* infection to uninfected bumblebees in the Czech Republic. The circles reflect the number of samples in sampling locations.

**Figure 3 insects-13-01121-f003:**
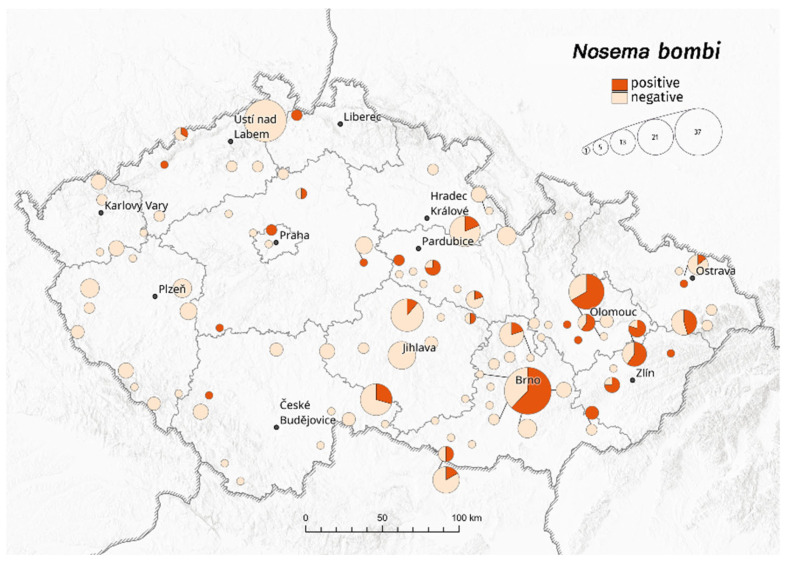
Sampling locations and the ratio of *Nosema bombi* infection to uninfected bumblebees in the Czech Republic. The circles reflect the number of samples in sampling locations.

**Figure 4 insects-13-01121-f004:**
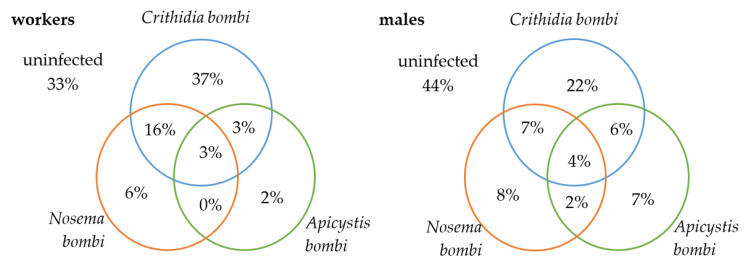
Percentage of individual PCR-positive *B. terrestris/lucorum* specimens individually and simultaneously infected with pathogens the *C. bombi*, *A. bombi* and *N. bombi,* especially for workers and males.

**Figure 5 insects-13-01121-f005:**
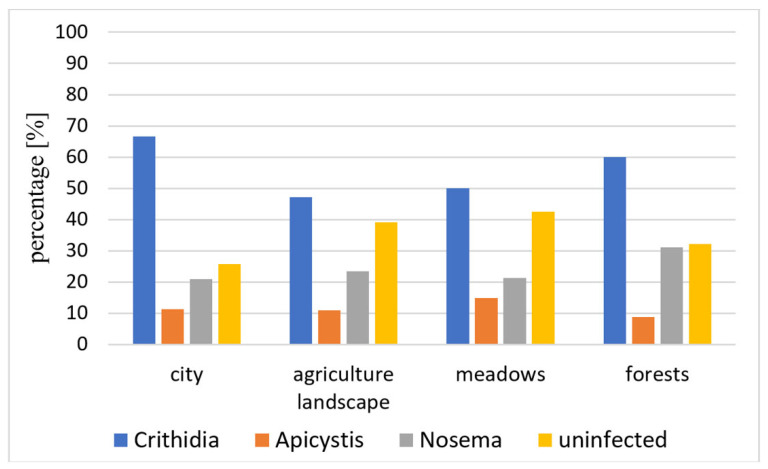
Presence of pathogens in *B. terrestris/lucorum* in various categories of landscape type independent of caste.

**Figure 6 insects-13-01121-f006:**
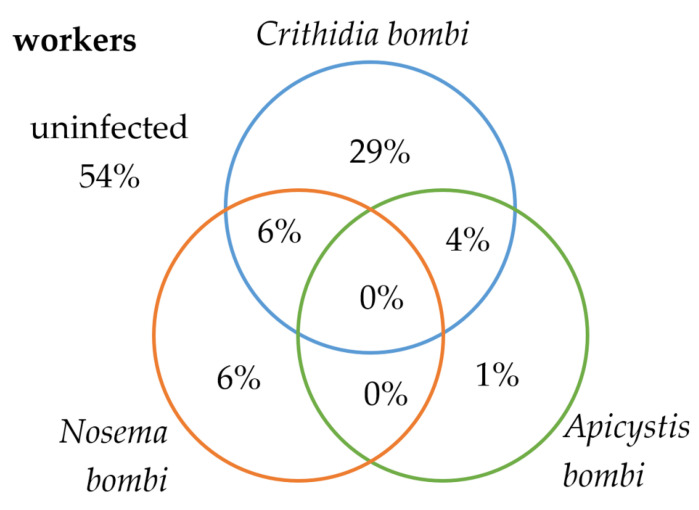
Percentage of individual PCR-positive *B. lapidarius* specimens individually and simultaneously infected with the pathogens *C. bombi*, *A. bombi* and *N. bombi* for workers.

**Table 1 insects-13-01121-t001:** Results of laboratory test, obtained by PCR detection for species *B. terrestris/lucorum* for pathogens *C. bombi*, *A. bombi* and *N. bombi*. W workers, M males, * low number of samples; - not evaluated.

*Bombus terrestris*/*lucorum*				*Crithidia bombi*				*Apicystis bombi*				*Nosema bombi*			
Year	Number of Locations	Workers	Males	W	%	M	%	W	%	M	%	W	%	M	%
2015	26	103	7	72	70	1	*	8	8	0	*	26	25	0	*
2016	6	64	5	41	64	4	*	11	17	0	*	23	36	2	*
2017	3	61	2	30	49	1	*	2	3	1	*	29	48	0	*
2018	18	68	38	35	51	20	53	9	13	16	42	6	9	8	21
2019	32	61	55	31	51	16	29	2	3	5	9	7	11	11	20
2020	41	39	37	23	59	15	41	2	5	5	14	8	21	9	24
Mean					57		41		8		22		25		22
Total	126	396	144	232		56		34		27		99		31	

**Table 2 insects-13-01121-t002:** Results of laboratory tests, obtained by PCR detection on the species *B. terrestris/lucorum* for pathogens *C. bombi*, *A. bombi* and *N. bombi*. W workers, M males. * low number of samples; - not evaluated.

*Bombus lapidarius*				*Crithidia bombi*				*Apicystis bombi*				*Nosema bombi*			
Year	Number of Locations	Workers	Males	W	%	M	%	W	%	M	%	W	%	M	%
2015	10	34	0	17	50	-	-	1	3	-	-	1	3	-	-
2016	6	31	7	9	29	3	*	1	3	1	*	3	10	1	*
2017	2	16	0	5	31	-	-	1	6	-	-	6	38	-	-
Mean					37				4				17		
Total	17	81	7	31		3		3		1		10		1	

## Data Availability

The data presented in this study are available on request from the corresponding author.

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
