# Peer review of "Prevalence and Distribution of Three Bumblebee Pathogens from the Czech Republic"

_insects, 2022, doi:10.3390/insects13121121_

Round 1

Reviewer 1 Report

In this study authors examined bumblebees for the presence of pathogens Crithida bombi, Apicystis bombi, and Vairimorpha (Nosema) bombi. The dependence of the spread of pathogens on the season and sex was investigated.

The manuscript is well-structured and the content is novel and deserves publication. 

I have only minor concerns:

15 line - it may be worth adding here the Latin names of the species "( buff-tailed bumblebee latin name and red-tailed bumblebee latin name)"

80-81 lines - at the first mention, the full name of the species shoul be given, then the abbreviated.

Please check the list of references, you have small errors

300 line - no comma

303 line

Author Response

Response to Reviewer 1 Comments

Point 1: 15 line - it may be worth adding here the Latin names of the species "( buff-tailed bumblebee latin name and red-tailed bumblebee latin name)"

Response 1: Corrected

Point 2: 80-81 lines - at the first mention, the full name of the species shoul be given, then the abbreviated.

Response 2: Corrected

Point 3: Please check the list of references, you have small errors

Response 3: Corrected

Reviewer 2 Report

I've found this research of great interest in order to know more about the health status of bumblebees populations. However, I have some comments for the authors to consider: 

L49-50: A reference is needed here to support this statement. 

Vairimorpha bombi: As far as I know, the change of name of Nosema to Vairimorpha has been suggested but not accepted yet. In any case, I think that at least should be noted that it is referring to the formerly known gender Nosema, and added the reference that suggest the name change.

L86 and 89: "...PCR diagnostic method with specific primers ... for the detection of A. bombi... according to Meeus et al. (2010)." But if one goes to that paper, it is named "Multiplex PCR detection of slowly-evolving trypanosomatids and neogregarines in bumblebees using broad-range primers". In the abstract, it is said: "A multiplex was designed containing an internal control and two broad-range primer pairs, detecting C. bombi and other SE trypanosomatids and also A. bombi and other neogregarines." And finally, in the material and methods: "A primer set Apicystis Universal =ApUF1: TCAATTGGAGGGCAAGTCTG ApUR1: CACGCAAAGTCCCTCTAAGAA (Fig. 1b) was designed". So the authors can not say that are using specific primers because it is not true. 

L120, 127 and 130: I consider better to include each figure after naming it to facilitate reading instead of putting the 3 of them together. 

L207-212: Here the authors discuss that the workers have more risk of infection, but I think that should be of interest to also address that males could be more susceptible than females to some pathogens and diseases due to their haploid condition, something that has been proved (O’Donnell and Beshers, 2004; Retschnig et al., 2014; Ruiz-González and Brown, 2006).

L246 and 247: I do not understand why the authors use the term "variant" to refer to coinfections of two pathogens.

L256: maybe cite some of these sub-lethal effects that could be mitigated would be of interest. 

Finally, I've detected some typos and spelling errors, so I suggest the authors to have a final check on that matter. 

Author Response

Response to Reviewer 2 Comments

Point 1: 15 line - it may be worth adding here the Latin names of the species "( buff-tailed bumblebee latin name and red-tailed bumblebee latin name)"

Response 1: Corrected

Point 2: L49-50: A reference is needed here to support this statement. 

Response 2: Corrected

Point 3: Vairimorpha bombi:  As far as I know, the change of name of Nosema to Vairimorpha has been suggested but not accepted yet. In any case, I think that at least should be noted that it is referring to the formerly known gender Nosema, and added the reference that suggest the name change.

Response 3: Corrected

Point 4: L86 and 89: "...PCR diagnostic method with specific primers ... for the detection of A. bombi... according to Meeus et al. (2010)." But if one goes to that paper, it is named "Multiplex PCR detection of slowly-evolving trypanosomatids and neogregarines in bumblebees using broad-range primers". In the abstract, it is said: "A multiplex was designed containing an internal control and two broad-range primer pairs, detecting C. bombi and other SE trypanosomatids and also A. bombi and other neogregarines." And finally, in the material and methods: "A primer set Apicystis Universal =ApUF1: TCAATTGGAGGGCAAGTCTG ApUR1: CACGCAAAGTCCCTCTAAGAA (Fig. 1b) was designed". So the authors can not say that are using specific primers because it is not true.

Response 4: Acording the text in the methods part we did not used universal primers combination ApUF1 / ApUR1 (Meeus, 2010) for Gregarinia which reviewer 2 was mentioned. But we used primer pair ApBF1 / ApUR2 (Meeus, 2010) from which ApBF1 is specific primer for A. bombi 18S and reverse primer ApUR2 is universal for 18S Gregarinia so the primers combination get A. bombi specific pcr product. In order to clarification method description we upgraded sentence on line 86-87: „Parasitic DNA was detected using a PCR diagnostic method with primers amplifying part 18S of rDNA specific for the studied parasites.“

Point 5: L120, 127 and 130: I consider better to include each figure after naming it to facilitate reading instead of putting the 3 of them together. 

Response 5: Corrected

Point 6: L207-212: Here the authors discuss that the workers have more risk of infection, but I think that should be of interest to also address that males could be more susceptible than females to some pathogens and diseases due to their haploid condition, something that has been proved (O’Donnell and Beshers, 2004; Retschnig et al., 2014; Ruiz-González and Brown, 2006).

Response 6: Thank you for an interesting topic for discussion. Our study was not designed to test ploidy difference hypotesis. There are strong genotype–genotype interactions between C. bombi and its Bombus hosts. Parasite transmission correlates with relatedness among hosts (Shykoff and Schmid-Hempel 1991), genetically diverse colonies suffer lower prevalence and intensity of infections (Baer and Schmid-Hempel 1999, 2001), patrilines differ in their susceptibility to the parasite (Baer and Schmid-Hempel 2003) and colonies differ in the strains of the parasite by which they can become infected (Schmid-Hempel et al. 1999; Mallon and Schmid-Hempel 2004; Schmid-Hempel and Reber Funk 2004). Furthermore, recent work shows that the parasite elicits an immune response in its host (Brown et al. 2003b) and that resistance to C. bombi depends on at least several loci and shows epistatic interactions (Wilfert et al. 2004). Ruiz-González and Brown, 2006 failed to prove this hypothesis under controlled conditions in the case of Crithidia bombi. We believe that more parameters should be monitored for a relevant discussion of this issue.

Point 7: L246 and 247: I do not understand why the authors use the term "variant" to refer to coinfections of two pathogens.

Response 7: Corrected

Point 8: L256: maybe cite some of these sub-lethal effects that could be mitigated would be of interest. 

Response 8: Corrected

Point 9: Finally, I've detected some typos and spelling errors, so I suggest the authors to have a final check on that matter. 

Response 9: Corrected